# Effects of jockey position and surfaces on horse movement asymmetry and horse-jockey synchronisation during trotting exercise

Kate Horan[1]*, Thilo Pfau[1,2]

**1** The Royal Veterinary College, Hertfordshire, United Kingdom, **2** University of Calgary, Calgary, Alberta, Canada

* khoran@rvc.ac.uk

## Abstract

Racehorses and jockeys can incur injuries, not only during gallops, but also during routine trotting exercise to access gallop tracks or warm-up. Understanding how jockey position affects racehorse movement may influence safety, and this may vary across different surface conditions. This study used inertial sensing technology (XSens MTw sensors) and linear mixed models to quantify and determine the significance ($p \leq 0.05$) of jockey riding position ('rising' versus 'two-point seat') and surface type (artificial, grass and tarmac) on: 1) time offsets between stance and flight phases; 2) horses' vertical upper body movement asymmetry and 3) time lags (in % of stride time) between horse and jockey maximum and minimum vertical displacements. Six ex-racehorses were recruited on a convenience basis from the British Racing School and were ridden by one jockey. Surface type did not significantly influence timings between the stance and flight phases or horse asymmetry. Jockey riding position was linked to a 1.8% difference in stance phase offsets ($p < 0.001$) and 0.9% difference in flight phases ($p = 0.015$) for two-point seat versus rising trot. Jockey riding position also affected horse movement asymmetry at the poll across stance phases (weight bearing asymmetry, $p = 0.005$) and symmetry at the withers and sacrum across flight phases (push-off asymmetry, $p < 0.001$). In rising trot, the jockey reduced poll asymmetry around the seated stance phase, but increased withers and sacrum push-off asymmetries after the seated stance phase. Time-offsets between the horse and jockey minimum and maximum displacements around stance and flight phases, respectively, were also significantly affected by jockey riding position (all $p < 0.001$). As the jockey stood up in their stirrups at stance, in either the rising component of rising trot or in the two-point seat, their delay in following the horse's movements increased by 2.8–4.5%, compared to when they were seated ($p < 0.001$). There was also an increased delay of the jockey by 0.6–0.8% around stance on tarmac compared to on the artificial surface ($p \leq 0.019$). During flight phases, jockey displacement maximums were reached 5.5–7.0% and 9.3–11% after the horse following

**Data availability statement:** All relevant data are within the manuscript and its Supporting information files. We have now also uploaded raw data to FigShare and provide a link in our manuscript.

**Funding:** This study was funded by a Horserace Betting Levy Board small grant (SPrj049) awarded to KH and TP. The funders had no role in the design of the study or the collection, analyses and interpretation of data.

**Competing interests:** I have read the journal's policy and the authors of this manuscript have the following competing interests: TP is the co-owner of Equigait, a provider of gait analysis products and services. This does not alter our adherence to all polices on sharing data and materials.

the seated stance in rising trot and during two-point seat, respectively, but jockey movements preceded horse movements around the post-standing flight phases by 4.9–7.3% in rising trot. In summary, jockey position had a greater impact on horse movement asymmetry and horse-jockey synchronisation than surface type. However, further work is required to relate study outcomes to injury risk.

## Introduction

Riding racehorses is a high-risk profession and optimising safety alongside performance is paramount. Aside from high-speed work under racing and training settings, it is also vital to understand how the safety and stability of both racehorses and their jockeys can be optimized when travelling over varied terrain at trot to access gallop tracks. This is particularly relevant for jockeys guiding young and inexperienced Thoroughbreds in training to tracks, and at sites that have already been identified as posing risks through horses slipping, such as tarmac roads. For example, Middleham in Yorkshire is a hotspot for racehorse and jockey injuries because it is the major racehorse training centre in northern England and this town experiences a very high volume of daily equestrian traffic. Over 500 racehorses are ridden daily along the roads in Middleham to get to and from their training gallops.

There are some studies that have investigated specific aspects of hoof-surface interactions on different surfaces [1–7] and ground surface type appears to be a significant risk factor for injuries to racehorses [8–13]. However, there are few studies that have investigated the potential implications of hoof-surface interactions on the upper body displacement patterns of racehorses and their jockeys [14]. Nevertheless, it is recognised that rider kinematics can adjust to accommodate changes in translational and rotational upper body movements of their horse [15], and vice versa [16,17]. Therefore, if hoof-ground interactions influence the upper body movement of the horse through alterations to the magnitude, direction, and origin of ground reaction force, they are expected to have relevance for horse-rider stability and movement efficiency. For example, excessive hoof slip may lead to biomechanical instabilities that prevent a horse from being able to support its centre of mass, thus causing it to stumble or fall, and potentially sustain an injury to its musculoskeletal system [18]. Even small adjustments at the shoe-surface interface have the potential to influence a horse's gait; for example, a unilateral hindlimb road nail, protruding 3 mm from a horseshoe surface was shown to induce small pelvic movement asymmetry in horses trotting on tarmac [19]. This effect was thought to reflect a reduction in hoof slip caused by the nail, and therefore increased time for weight-bearing and generation of propulsive force during stance. Jockey injuries are very often linked to horse falls [20], therefore factors that influence a horse's gait may have repercussions for the risk of jockey injury and falls. This may be particularly catastrophic if horse slipping incidents coincide with passing vehicular traffic.

In addition to hoof kinematics, the riding position adopted by the jockey might also be a relevant consideration for optimising safety, as a jockey's weight distribution

may affect both their own stability and that of their horse. In sitting trot, a horse and rider should achieve a high degree of coupling [21–23], while in a crouched two-point seat position a rider's body is less tightly coupled to the movements of their horse [24–26].

The purpose of the present study was to assess racehorse-jockey interactions over three surface types (artificial, grass and tarmac) at trot, with the jockey trialling rising and two-point seat riding positions. The surfaces selected were expected to initiate fundamental differences in hoof kinematics (e.g., slip distance) and 'stride to stride' variation: tarmac (hard and consistent); grass (variably hard and inconsistent); artificial (soft and intermediate consistency). We hypothesised that if hoof placement and orientation on landing vary more on softer surfaces, such as artificial and grass compared to tarmac [27], then the upper body movements of the horse and jockey may also become less well constrained on these softer surfaces. We hypothesised that there would be a greater offset in time between horse and jockey movements when the jockey adopts either a two-point seat position or is standing during rising trot, compared to when they are seated in the saddle during rising trot. We also hypothesised that the jockey would impact the movement asymmetry of the horse, particularly during rising trot when the jockey imposes an uneven biphasic load over the course of the trot stride cycle.

## Materials and methods

### Ethics

Ethical approval for this study was received from the Royal Veterinary College Clinical Research Ethical Review Board (URN 2020 2001-2), which included a written consent form for horse owners and jockeys.

### Horse and rider participants

A convenience sample of six retired Thoroughbred racehorses in regular work and utilised for jockey education at the British Racing School (BRS) in Newmarket, UK, was included in this study. These horses are no longer involved in competitive racing but are fit and work under conditions similar to those encountered during active race training. The horses had ages between 7 and 19 years, their masses ranged from 510 to 580 kg, and their heights ranged from 16 to 17 hh (1.63 to 1.73 m). The same jockey rode all horses and had a body mass of approximately 60 kg. The jockey had been chosen to ride the horses in a racing exercise saddle on 22/05/2021 by the BRS management team. They have a category A and point-to-point license and approximately ten years of experience in the racing industry. They are riding racehorses daily and as such were fit to participate in multiple trials with all horses and they did not become fatigued during the data collection.

### Equipment

Six inertial measurement unit (IMU) devices (XSens MTw), each with a mass of 16 g, were positioned at the poll, withers, left and right tuber coxae, first lumbar vertebra (L1), and sacrum of the horses, as part of a validated sensor-based system [28,29]. In addition, three IMUs were fitted to the pelvis, mid- and upper-back of the jockey. The same researcher identified the anatomical positions on the horse and jockey by palpation and then secured the devices using custom-made Velcro pouches and doubled-sided tape. Time-synchronized IMU data were collected with an update rate of 60 Hz per individual sensor channel and transmitted to a receiver station (Awinda, Xsens) via a proprietary wireless data transmission protocol (XSens), which was connected to a laptop computer (Windows 11 Pro, 64-bit operating system, 16 GB RAM, Intel(R) Core (TM)i7-10610U CPU processor, 512 GB disk space) running MTManager (XSens) software.

### Trial conditions

During trotting exercise, jockeys either move up and down, in and out of the saddle, in time to the stance phases of diagonal pairs of limbs ('rising trot') or remain standing ('two-point seat') during each stride cycle. In this study, the horse-jockey

dyads performed ridden trials in both rising trot and two-point seat positions to facilitate a comparison of ridden states with differing amounts of expected horse-rider coupling. Trials were performed on tarmac, an artificial training track (Martin Collins Activ Track) and a grass training track. For the rising trot condition, there were two types of trials: one with the jockey sitting in the saddle during the left forelimb stance and rising out of the saddle during right forelimb stance (defined as rising trot – left diagonal) and one with the jockey sitting in the saddle during the right forelimb stance and rising out of the saddle during the left forelimb stance (rising trot – right diagonal). The artificial surface was deemed to be of 'standard' going and the grass was 'good to firm' during data collection. In addition to the ridden trials on the artificial surface, grass, and tarmac, to acquire reference movement symmetry values, horses were additionally trotted in-hand in a straight line over the tarmac wearing a bridle, with the handler on the horse's left side. Conscious effort was made to ensure the horses' heads were unrestrained during this in-hand trotting exercise. The distribution of stride times across trial conditions is provided in S1 Fig in S1 File.

### Data processing and statistics

**Overview.** The IMU data were processed following published protocols [28]. In brief, the tri-axial sensor acceleration data were rotated into a gravity (z: vertical, positive up) and horse-based (x: cranio-caudal, positive forwards; and y: medio-lateral, positive to the left) right-handed reference frame and numerically double integrated to displacement. Displacement data were then segmented into individual strides based on the vertical velocity of the sacrum sensor [30]. The data for each stride were linearly interpolated to 100 time percentage points and the average displacement across strides within a trial was calculated at each time point for each of the x, y, z axes of each sensor. On average, 23 strides were used to calculate average displacement. From the displacement data, the following three outcome parameters were evaluated.

**Time between stance and flight phases.** There are two stance phases and two flight phases per trot stride, reflecting the weight support and push-off phases of diagonal pairs of limbs (right fore and left hind; left fore and right hind). The time points (in percent of the total stride) of the two displacement minima (representing mid-stance) and maxima (representing mid-flight) for the horse were noted for each trial and time offsets (in percent stride time) were calculated (Fig 1). An averaged set of displacement data for horse movement at the withers and L1 ('withers and L1') was used to identify the time points of the minima and maxima. This is because the data from the 'withers and L1' landmark was also used in the evaluation of time-lags between horse and jockey movements (see section 'Time lags between vertical horse and rider upper-body displacements' below).

Initially, we calculated offsets as time at 'stance 2 – stance 1' and time at 'flight 2 – flight 1'. However, to simply compare between rising and two-point seat conditions, data for the rising trot on both diagonals were pooled; this was achieved by subtracting the time offsets for data from the left diagonal from 100%. This means that the stance time offset for the rising data represents the time between the stance phase when the jockey is seated through to the stance phase when they are standing (time point when standing in stance minus time point when seated in stance). The flight time offset for the rising trot data represents the time between the flight phase after the jockey is seated through to the flight phase after they were standing (flight time point when standing after standing stance minus flight time point when standing after the seated stance).

**Horse movement symmetry.** To evaluate the gait symmetry of the horses under the different riding positions and on different surfaces, differences between minima, maxima and upwards amplitudes (MinDiff, MaxDiff and UpDiff, respectively) were calculated [31] at the poll, withers and sacrum. The differences between left and right-sided displacement were calculated such that, in all cases, positive values for MinDiff, MaxDiff and UpDiff indicated an increase in right-sided asymmetry (i.e., an indication of reduced force production during right limb stance [32,33]) at either the poll, withers or sacrum.

To deal with the horses' underlying asymmetries, we considered the data for the horses trotting in-hand on hard level tarmac to be representative of their 'baseline' asymmetry. If MinDiff, MaxDiff or UpDiff parameters indicated a 'left

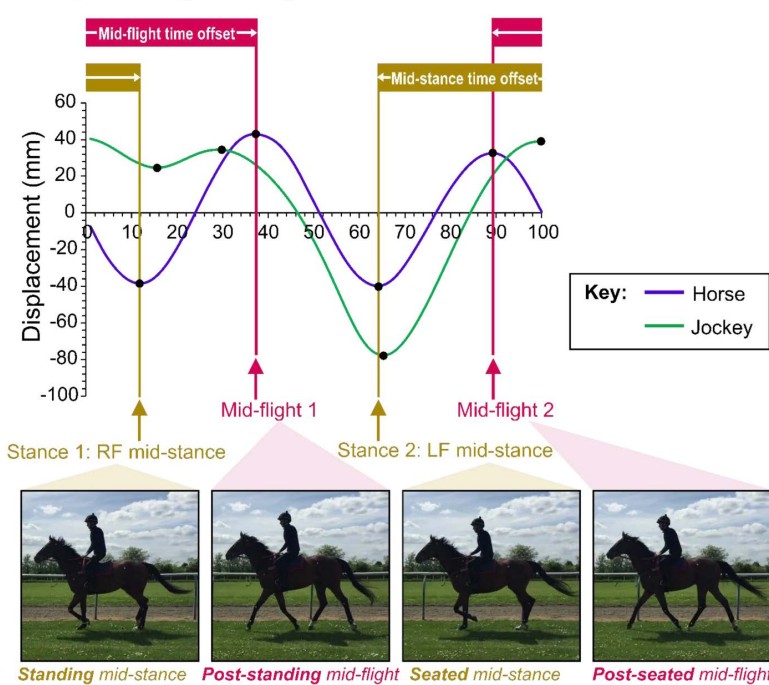

**Fig 1. Illustration of vertical displacement minima and maxima for the horse (and jockey) over a stride cycle indicating the stance and flight time offsets calculated.** Example shows data for the rising trot left diagonal when the jockey was seated during left forelimb stance. Time along the x-axis is given in percent stride time. RF = right forelimb. LF = left forelimb.

asymmetry' (i.e., a pattern consistent with reduced force production of the left limb) all data for that horse for the parameter in question were inverted (multiplied by -1) and the diagonal relabelled. Therefore, after implementing this procedure, all horses displayed baseline movement asymmetries consistent with reduced force production through the right limb ('right asymmetrical' movement). The resulting MinDiff, MaxDiff and UpDiff values for 'baseline' were then subtracted from the resulting values for the different conditions to reveal surface and riding position effects.

**Time lags between vertical horse and jockey upper-body displacements.** The time points of the two vertical displacement minima and maxima for both the horse and jockey were determined for each trial and the time offsets (in % of the stride cycle) calculated. Data from the trials with the jockey rising on the left diagonal were swapped around so the 'rising trot' data could be pooled together. As such, the following parameters for the rising trot data were calculated:

- **StanceTDiff – seated**: the time lag between horse and jockey displacement minima around the horse's forelimb stance that corresponds to the diagonal name. When the jockey was on the 'right diagonal' this is the difference between the vertical horse and jockey displacement minima in the first half of the stride cycle (Fig 2A); when the jockey was on the 'left diagonal' this is the difference between the horse and jockey displacement minima in the second half of the stride cycle (Fig 2B).

- **StanceTDiff – standing**: the time lag between horse and jockey displacement minima around the horse's forelimb stance that does not correspond to the diagonal name. When the jockey was on the right diagonal this is the difference between the horse and jockey displacement minima in the second half of the stride cycle (Fig 2A). When the jockey was on the left diagonal this is the difference between the horse and jockey displacement minima in the first half of the stride cycle (Fig 2B).

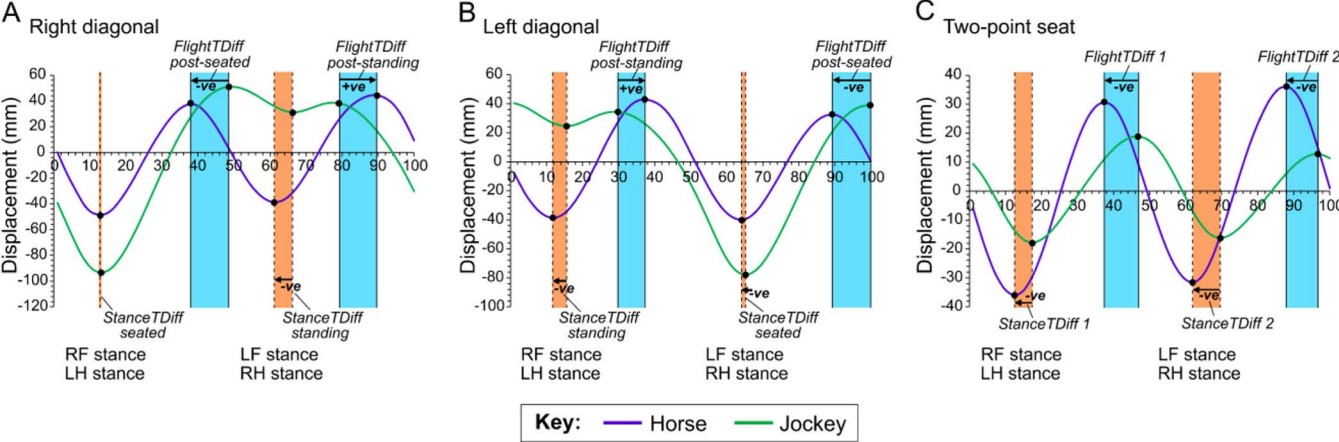

**Fig 2. Illustration of vertical displacement minima and maxima for the horse and jockey over a stride cycle. A)** Jockey on the right diagonal. **B)** Jockey on the left diagonal. **C)** Jockey in the two-point seat position. The timing of the stance phases is indicated for horse's right-forelimb (RF), left forelimb (LF), right hindlimb (RH) and left hindlimb (LH). Examples show data collected from Horse 1 on the artificial surface.

- **FlightTDiff – post-seated**: the time lag between the most upward displacement point of the horse and jockey during the flight phase after seated stance. When the jockey was on the 'right diagonal' this is the difference between the horse and jockey displacement maxima in the first half of the stride cycle (Fig 2A). When the jockey was on the 'left diagonal' this is the difference between the horse and jockey displacement maxima in the second half of the stride cycle (Fig 2B).

- **FlightTDiff – post-standing**: the time lag between the most upward displacement point of the horse and jockey during the flight phase after standing stance. When the jockey was on the right diagonal this is the difference between the horse and jockey displacement maxima in the second half of the stride cycle. When the jockey was on the left diagonal this is the difference between the horse and jockey displacement maxima in the first half of the stride cycle (Fig 2A and 2B).

In the two-point seat position, there is no systematic reason to expect differences between the minimum (StanceTDiff) and maximum (FlightTDiff) displacement values of the horse and jockey in the two halves of the stride cycle, therefore these data were pooled (Fig 2C).

The inertial sensors that were positioned at the withers and first lumbar vertebra were closest to the jockey, therefore the time offsets between the jockey's displacement (at the pelvis, mid-back and upper-back) and the horse's displacement, averaged across 'withers and L1', were assessed.

## Statistics

Linear mixed models were implemented in IBM SPSS statistical software (version 28) to assess whether surface, jockey riding position, stride time or two-way interactions between these parameters (fixed factors) significantly affected the time between horse stance and flight phases of the horse, horse asymmetry parameters or time lags between horse and jockey displacements. Horse was included as a random factor in the models, as we were not interested in the effects of individual horses. Stride time was included as a covariate (as well as a fixed factor) because we were also interested in its numerical relationship with the outcome parameters; for example, stride time is typically related to speed [34] and it seemed important to establish whether the investigated variables would increase or decrease with speed (stride time). The significance threshold was set at $p < 0.05$. In all cases, models adopted the Restricted Maximum Likelihood method

and the Satterthwaite approximation for the degrees of freedom. The p value outputs for the interaction terms of these initial linear mixed models were evaluated. If any p values for interaction terms exceeded 0.1 then these terms were removed, and final models were run with a reduced number of fixed terms to lower statistical noise. We chose this method in agreement with previous studies [e.g., 5,18]. This meant that the single fixed factors were reevaluated alongside only the interaction terms that had been significant in the preliminary models. In each case for the final models, histograms of model residuals were plotted and visually inspected for normality; no data transformations were deemed necessary. All post-hoc pairwise comparisons included Bonferroni correction.

## Results

### Time offsets between the horses' stance and flight phases

These data are illustrated in Fig 3. The preliminary linear mixed model to investigate timings between stance and flight phases indicated that all interaction terms had high p values (p ≥ 0.132). The final models therefore only included the main effects (surface, jockey riding position and stride time) in each case. Only jockey riding position had a significant effect on the time offsets between the stance (p < 0.001) and flight (p = 0.015) phases. For the stance phase time offsets, there was a 1.8% (1.0–2.6%; 95% confidence interval for the difference) smaller time offset for the rising trot condition compared to the two-point seat position. This indicates that the stance phases became closer together after the 'seated' portion of the rising trot (see Fig 1 for illustration of calculated time differences). For flight phase time offsets, there was a 0.9% (0.2–1.6%; 95% confidence interval for the difference) lower time offset for the rising trot condition compared to the two-point seat position. Here, this indicates that the flight phases became slightly closer together after the 'post-seated' portion of the rising trot (Fig 1). All significance values and estimated marginal means for surface, jockey riding position and surface plus jockey position effects on the stance and flight time offsets are provided in the Supplementary materials (S1–S4 Tables in S1 File).

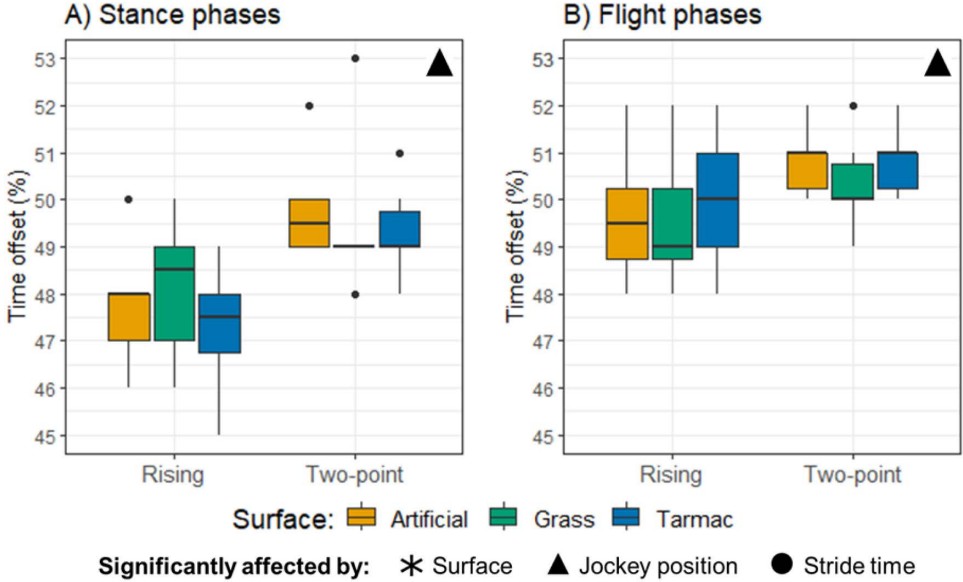

**Fig 3. Time offsets between the A) stance and B) flight phases.** Data are sub-divided according to jockey riding position (rising trot or two-point seat) and coloured according to surface type.

## Horse movement symmetry

**Poll.** MinDiff: The preliminary linear mixed model indicated that the surface* jockey riding position interaction term had a high p value (p = 0.334). The final model included the main effects (surface, jockey riding position and stride time), as well as surface*stride time and the jockey riding position*stride time two-way interactions. In this final model, jockey riding position (p = 0.005) and jockey riding position*stride time (p = 0.008) were found to have significant effects. All jockey riding positions were associated with negative values, indicating left-sided asymmetry, but this was most noticeable for the right diagonal, which had an estimated marginal mean of -16 mm. The left diagonal had an estimated marginal mean of -1.8 mm, and the two-point seat had an estimated marginal mean of -9.5 mm (Fig 4).

MaxDiff: The preliminary mixed model indicated that all interaction terms had high p values (p ≥ 0.213), so they were excluded from the final model. In the final model, neither surface, jockey riding position nor stride time had significant effects (p ≥ 0.187).

UpDiff: The preliminary mixed model indicated that all interaction terms had high p values (p ≥ 0.487). In the final model neither surface, jockey riding position nor stride time had significant effects (p ≥ 0.189).

**Withers.** MinDiff: The preliminary linear mixed model indicated that the surface*jockey riding position interaction term had a high p value (p = 0.939). In the final model, including the main effects (surface, jockey riding position and stride time) and the surface*stride time and jockey riding position*stride time interaction terms, surface (p = 0.042), jockey riding position (p = 0.021), surface*stride time (p = 0.049) and jockey riding position*stride time (p = 0.023) were all significant. However, the post-hoc pairwise comparisons with Bonferroni correction did not identify any pairwise significant differences amongst the surfaces or jockey riding positions.

MaxDiff: The preliminary mixed model indicated that surface*jockey riding position and surface*stride time interactions had high p values (p ≥ 0.478) and they were therefore excluded from the final model. In the final model, both jockey riding position (p < 0.001) and jockey riding position*stride time (p = 0.008) were found to have significant effects on the MaxDiff parameter. All jockey riding positions were significantly different to one another (p < 0.001). The right diagonal was associated with a positive asymmetry value (12.5 mm), representing right-sided asymmetry, whereas the left diagonal was associated with a negative asymmetry value (-9.8 mm), representing left-sided asymmetry. The asymmetry value for the two-point seat position fell in the middle, with a value close to zero (0.6 mm).

UpDiff: The preliminary mixed model indicated that both surface*jockey riding position and surface*stride time had high p values (p ≥ 0.338). The final model, excluding these interaction terms, indicated that jockey riding position (p = 0.001) and jockey riding position*stride time (p = 0.004) were significant, but all other p values were ≥ 0.323. All jockey riding positions were significantly different to one another (p < 0.001). The right diagonal was associated with a positive asymmetry value (7.9 mm), representing right-sided asymmetry, whereas the left diagonal and two-point seat positions were associated with negative asymmetry values (-14.6 mm and -3.4 mm, respectively), representing left-sided asymmetry.

**Sacrum.** MinDiff: The preliminary linear mixed model indicated that all interaction terms had high p values (p ≥ 0.234), so they were excluded from the final model. The final model indicated that neither surface, jockey riding position nor stride time had significant effects (p ≥ 0.230)

MaxDiff: The preliminary mixed model indicated that all interactions had high p values (p ≥ 0.326), so they were all removed from the final model. The final model indicated that surface (p = 0.031) and jockey riding position (p < 0.001) had significant effects. However, the post-hoc pairwise comparisons did not highlight significant differences amongst the surfaces. The post-hoc comparisons did reveal that all jockey riding positions were significantly different to one another (p < 0.001), with the right diagonal rising trot position being associated with a negative asymmetry value (-9.4 mm) indicating left-sided asymmetry in the hind end, whereas the left diagonal was associated with positive asymmetry (11.0 mm), reflecting increased right-sided asymmetry. The two-point seat position had a MaxDiff asymmetry value of 2.0 mm.

UpDiff: The preliminary mixed model indicated that all interactions had high p values (p ≥ 0.177). The final model, excluding these interaction terms, indicated that jockey riding position (p < 0.001) and stride time (p = 0.010) had significant

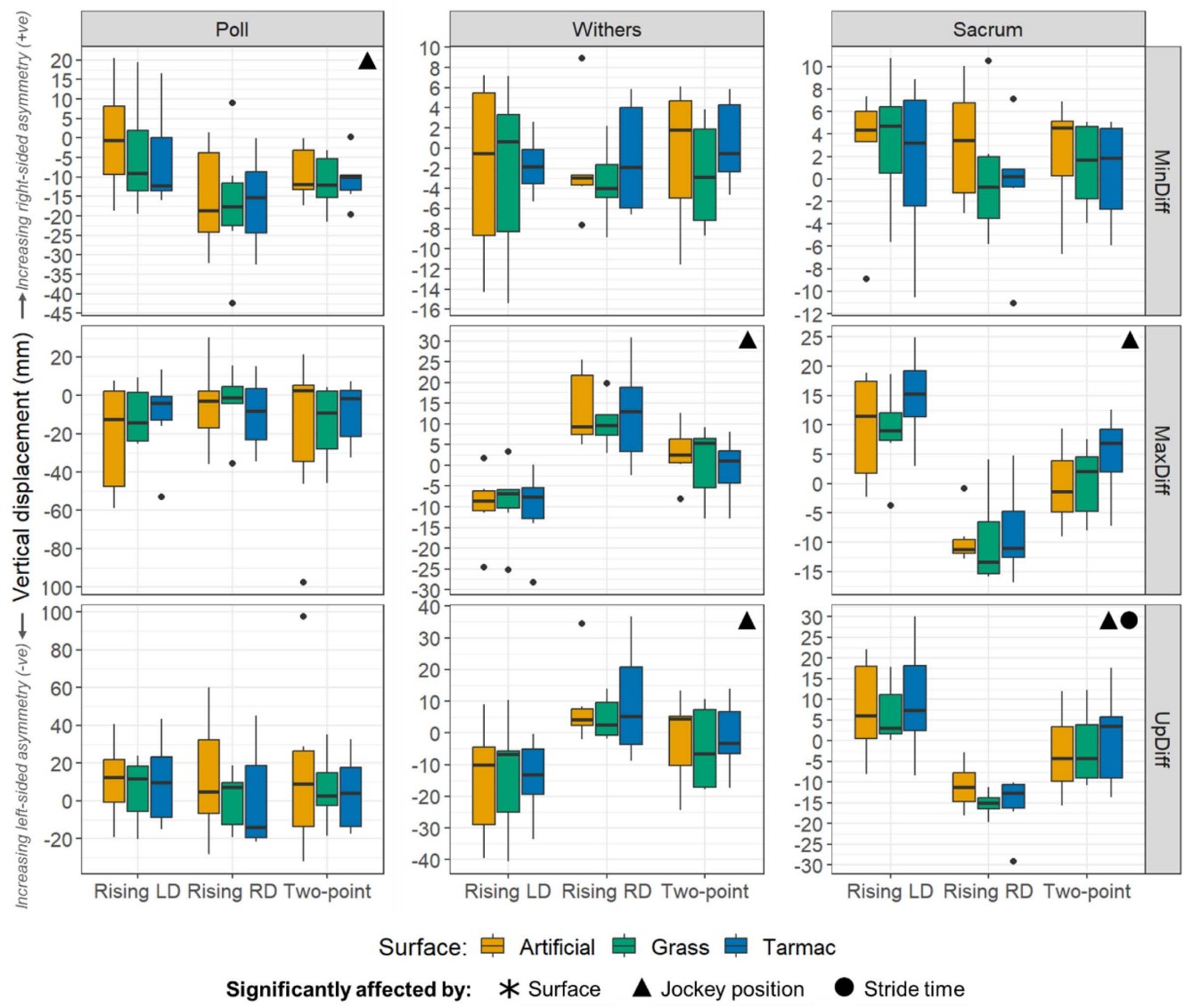

**Fig 4. MinDiff, MaxDiff and UpDiff asymmetry parameters at the poll, withers, and sacrum.** The horses' baseline asymmetry data (unridden trot on tarmac) were subtracted from the data presented to reveal surface and jockey riding position effects. Boxplots are subdivided according to jockey riding position and coloured by surface. Positive values indicate right-sided asymmetry; negative values represent left-sided asymmetry. In cases where the linear mixed models suggested a significant effect of either surface or jockey riding position but the post-hoc pairwise comparisons did not identify any significant differences amongst the specific surface or jockey riding position categories, then the asymmetry parameter (MinDiff, MaxDiff or UpDiff) has not been marked as 'significantly affected'. LD = left diagonal. RD = right diagonal. MinDiff = difference between displacement minima. MaxDiff = difference between displacement maxima. UpDiff = Difference between upwards amplitudes.

effects. All jockey riding positions were significantly different to one another (p < 0.001 in each comparison). The right diagonal was associated with a negative asymmetry value (-14.7 mm), reflecting left-sided asymmetry, whereas the left diagonal was associated with an asymmetry value of 7.8 mm, representing right-sided asymmetry. The asymmetry value for the two-point seat position was -0.3 mm, indicating that the movement was almost symmetrical. Fig 4 summarises the MinDiff, MaxDiff and UpDiff results for the poll, withers, and sacrum.

The relationship between the asymmetry parameters and stride time is summarised in Fig 5.

All significance values and estimated marginal means for surface, jockey riding position and surface plus jockey riding position effects on the asymmetry parameters are provided in the Supplementary materials (S5–S8 Tables in S1 File).

## Time lags between vertical horse and jockey upper-body displacements

**Displacement minimum offsets (horse mid-stance).** At the jockey's pelvis, the preliminary linear mixed model used to investigate time offsets between horse and jockey minimum displacements indicated that surface*jockey riding position and surface*stride time had high p values (p = 0.455 and 0.209, respectively). In the final model, including the main effects (surface, jockey riding position and stride time) and jockey riding position*speed, all factors were found to be significant (p ≤ 0.005). Amongst the surfaces, artificial and tarmac were found to be significantly different (p = 0.008), with the latter associated with a reduced time lag of 0.7%. For the jockey riding positions, StanceTDiff-seated,

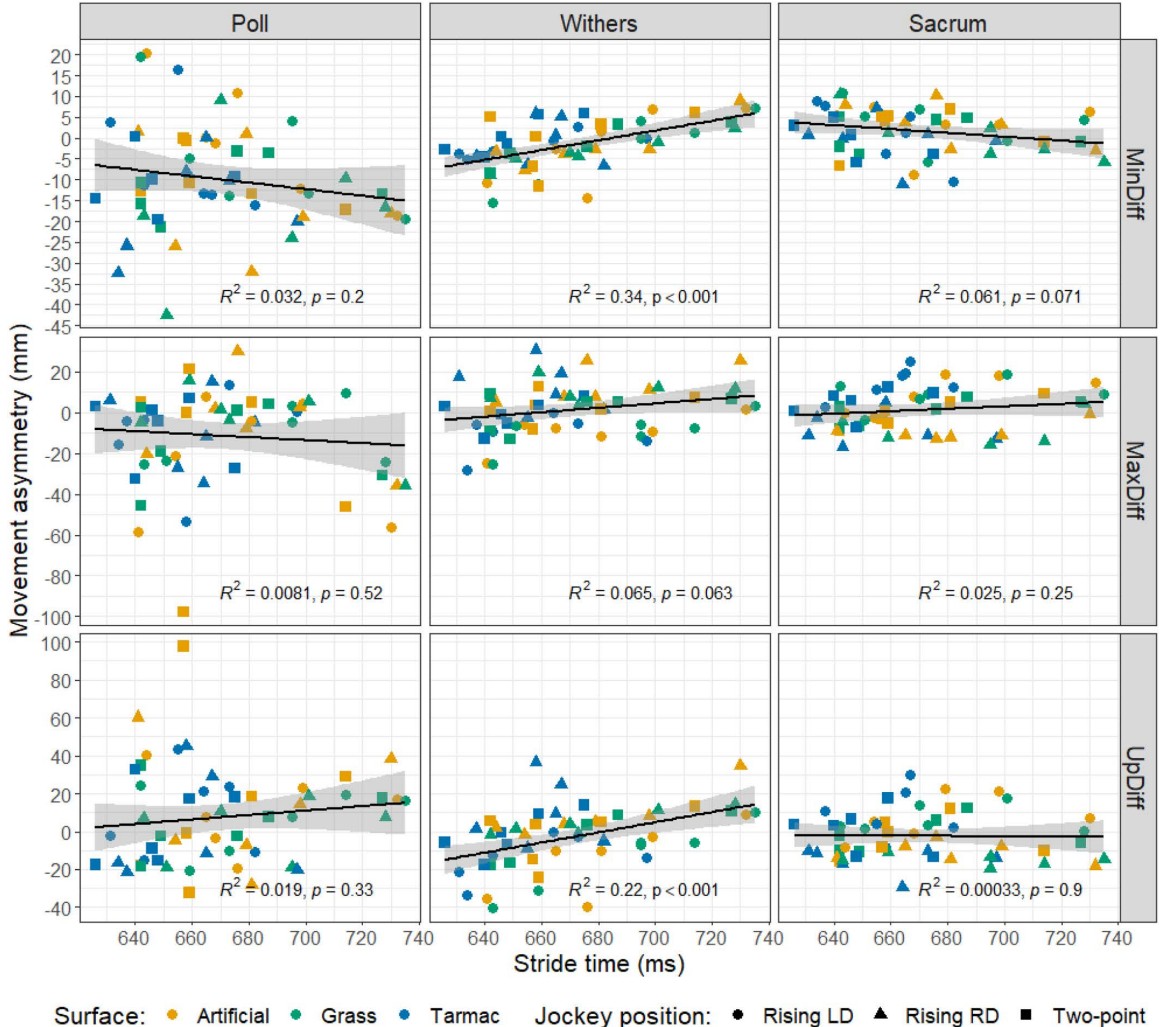

**Fig 5. Relationship between movement asymmetry parameters and stride time at the poll, withers, and sacrum.** Data are coloured according to surface type and symbol shape represents jockey riding position. LD = left diagonal. RD = right diagonal. $R^2$ and p values are indicated on each plot.

StanceTDiff-standing and Stance TDiff for two-point seat were all significantly different to one another (p < 0.001). The time lag for the jockey's pelvic movement to follow the horse and reach the minimum was 4.5% and 3.4% longer for StanceTDiff-standing and StanceTDiff for two-point seat, respectively, compared to StanceTDiff-seated. The time lag for the jockey was 1.1% shorter for StanceTDiff-standing compared to StanceTDiff for two-point seat.

At the jockey's mid-back, the preliminary linear mixed model indicated that all interaction terms had high p values (p ≥ 0.160). In the final model, including only the main effects (surface, jockey riding position and stride time), all factors were found to be significant: surface (p = 0.024), jockey riding position (p < 0.001) and stride time (p < 0.001). Amongst the surfaces, artificial and tarmac were found to be significantly different (p = 0.019), with the latter associated with a reduced time lag of 0.6%. For the jockey riding positions, StanceTDiff-seated and StanceTDiff-standing were significantly different (p < 0.001), with the StanceTDiff-seated associated with 3.9% less delay of the jockey. StanceTDiff-seated and the StanceTDiff for the two-point seat were also significantly different to one another (p < 0.001), with the former associated with a 3.5% lower time lag for the jockey.

At the jockey's upper-back, the preliminary linear mixed model indicated high p values for the surface*jockey riding position (p = 0.555) and surface*stride time (p = 0.312) interaction terms. In the final model, including the main effects (surface, jockey riding position and stride time) and jockey riding position*stride time, all factors were found to be significant (p ≤ 0.005). Again, amongst the surfaces, it was the artificial and tarmac that were found to be significantly different (p = 0.004), with the latter in this instance associated with a reduced time lag of 0.7% for the jockey to reach its minimum displacement position after the horse. The other surface comparisons were non-significant. For the jockey riding positions, all were found to result in significantly different time lags to one another (p < 0.001). The time lag for the jockey's pelvic movement to follow the horse was 2.8% and 4.0% greater for StanceTDiff-standing and two-point seat, respectively, compared to StanceTDiff-seated. The two-point seat StanceTDiff was also associated with 1.1% reduced delay compared to StanceTDiff-standing, in contrast to the lower back. The trends for the time offsets between the horse and jockey minima are illustrated in Fig 6A.

**Displacement maximum offsets (horse mid-flight).** At the jockey's pelvis, the preliminary linear mixed model to investigate time offsets between horse and jockey maximum displacements indicated that surface*jockey riding position and surface*stride time had high p values (p = 0.122 and p = 0.845, respectively). In the final model, including the main effects (surface, jockey riding position and stride time) and jockey riding position*speed, all factors were found to be significant (p ≤ 0.038). However, the differences amongst surfaces were small and the post-hoc analysis did not reveal any significantly different pairwise comparisons. For the jockey riding positions, FlightTDiff post-seated, FlightTDiff post-standing and two-point seat were all significantly different to one another (p < 0.001). The jockey's pelvic movement preceded the horse for FlightTDiff post-standing, resulting in a difference of 14.2% and 10.4% compared to FlightTDiff post-seated and FlightTDiff in two-point seat. The lag for the jockey to reach maximum displacement after the horse in two-point seat was also 3.8% less than for FlightTDiff post-seated.

At the jockey's mid-back, the preliminary linear mixed model to investigate time offsets between horse and jockey maximum displacements indicated that the surface*stride time interaction term had a high p value (p = 0.628), so this was excluded from the final model. In the final model, jockey riding position (p < 0.001) and jockey riding position*stride time (p < 0.001) were found to be significant, but surface, stride time and surface*jockey riding position were non-significant (p ≥ 0.054). For the jockey riding positions, all conditions were significantly different (p < 0.001), but the most notable differences were between FlightTDiff post-standing and FlightTDiff post-seated (15.7%) and FlightTDiff post-standing and FlightTDiff for two-point seat (11.7%), with FlightTDiff-post-standing again being associated with a positive time offset (jockey ahead of horse), whereas the FlightTDiff post-seated and FlightTDiff for two-point seat were associated with a negative time lag, with the jockey following the movement of the horse in reaching their peak vertical displacement. The lag for FlightTDiff in two-point seat was also 4.0% less than for FlightTDiff post-seated.

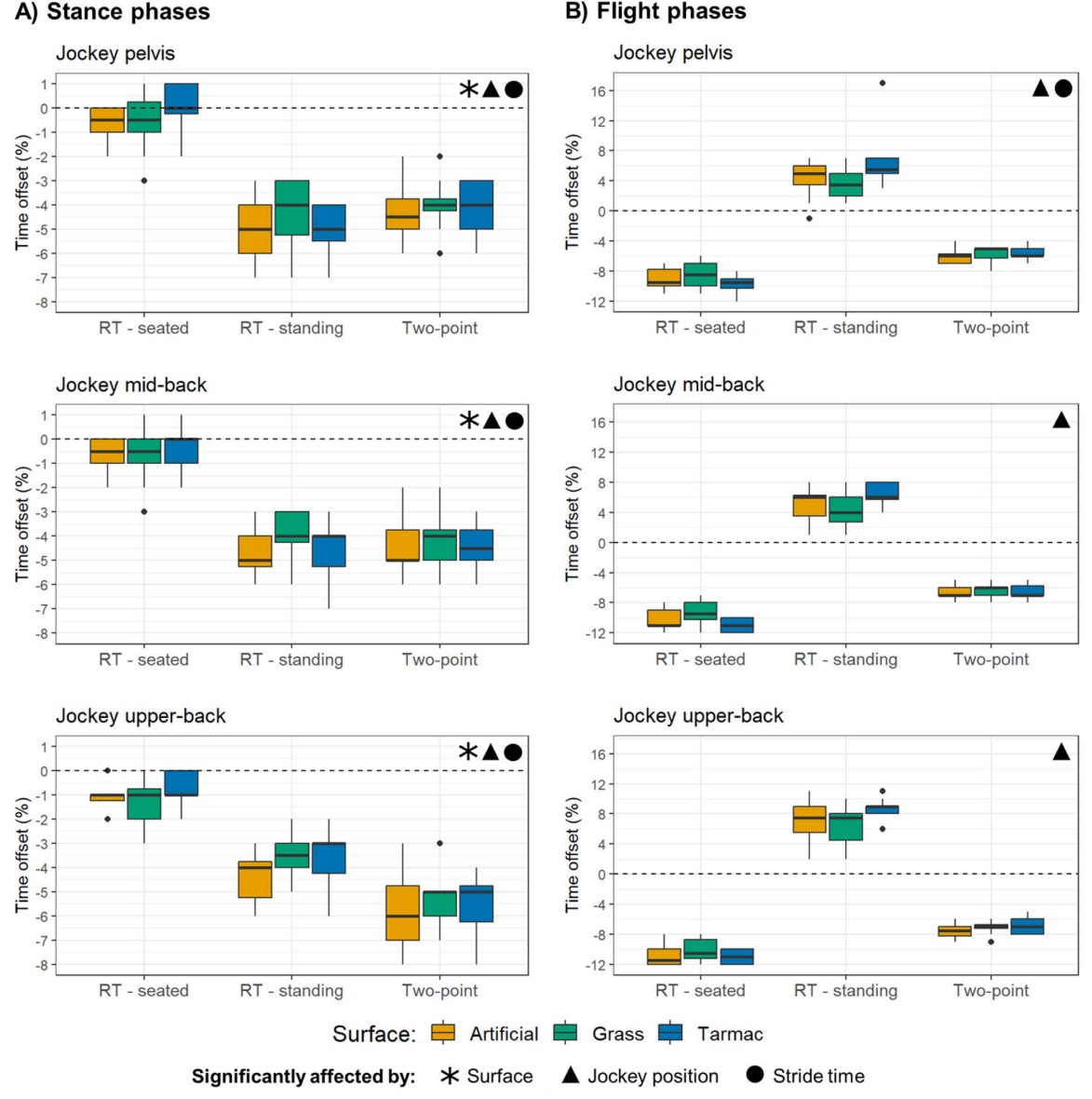

**A) Stance phases**

**B) Flight phases**

Surface: ■ Artificial ■ Grass ■ Tarmac

**Significantly affected by:** ✳ Surface ▲ Jockey position ● Stride time

**Fig 6. Time offsets between the horse-jockey movements at A) stance phases (minimum displacements) and B) flight phases (maximum displacements).** Comparisons were made between the jockey pelvis, mid-back and upper-back versus the 'withers and L1 landmark' on the horse. In cases where the linear mixed models suggested a significant effect of either surface or jockey riding position but the post-hoc pairwise comparisons did not identify any significant differences amongst the specific surface or riding position categories, then the time offset has not been marked as 'significantly affected'. RT = rising trot.

At the jockey's upper-back, the preliminary linear mixed model to investigate timing offsets between horse and jockey minimum displacements indicated that surface*jockey riding position and surface*stride time had high p values (p = 0.242 and 0.523, respectively). In the final model, including the main effects (surface, jockey riding position and stride time), jockey riding position and jockey riding position*stride time were significant (p < 0.001); all other p ≥ 0.065. Here, it was also the case that for the jockey riding positions, all conditions were significantly different (p < 0.001), and the most notable differences were again between FlightTDiff-post-standing and FlightTDiff-post-seated (18.3%) and

FlightTDiff-post-standing and FlightTDiff for two-point seat (14.3%), with FlightTDiff-post-standing being associated with a positive offset, whereas the FlightTDiff-post-seated and FlightTDiff in two-point seat were associated with a negative time lag (jockey following the horse movement). The trends for the time offsets between the horse and jockey minima are illustrated in Fig 6B.

All significance values and estimated marginal means for surface, jockey riding position and surface plus jockey riding position effects on the time lags between horse and jockey displacements are provided in the Supplementary materials (S9–S12 Tables in S1 File).

## Discussion

Quantification of surface characteristics, in addition to horse and rider upper body movement parameters, appears to be of specific importance in the context of musculoskeletal performance, injury, lameness and days lost from training/competition [2,14,35–37]. The present work delivered baseline data on ex-racehorses' trot kinematics, including time intervals between their displacement minima and maxima (occurring at the mid-stance and mid-flight phases, respectively), and movement asymmetry, and racehorse-jockey coordination dynamics, as a jockey adopted two different riding positions, whilst trotting over tarmac, an artificial racing track and grass. The surfaces selected are common types that racehorses and their jockeys pass over to access gallop tracks, they are routinely used for warming-up, and two of them are also used during training and racing (artificial and grass). The results from this study will be applicable to different racehorse training centres. They are also relevant to riders in other disciplines and may have wider implications for the welfare of horses and jockeys in both rural and urban areas; for example, in areas where there is a high level of horse traffic or horse-drawn vehicles, as well as in instances where horses are ridden on roads and different surfaces for leisure purposes.

### Surface

Overall, the effect of surface on the time offsets between mid-stance and mid-flight phases, horses' vertical upper body movement asymmetry, and the time lags between horse and jockey maximum and minimum vertical displacements was limited compared to jockey riding position effects. It is logical that surface had no effect on the time offsets between the two stance phases and time offsets between the two flight phases, as any alterations to stride timing variables would be expected to have a comparable impact on the two halves of the stride cycle. In addition, although there may be differences in horse upper body range of motion at trot on different surface types (Horan et al., in prep.), the fact that there was a consistent bilateral change in the hoof-ground interaction in the two halves of the trot cycles may also explain why there was little change in asymmetry. MinDiff values at the withers and sacrum initially appeared to be impacted by surface based on model outputs, but the post-hoc pairwise comparisons indicated that there were no significant differences amongst the surface types. For comparison, in a previous study, which quantified movement asymmetry in horses trotting over different surfaces, it was observed that there was a significant difference in poll MaxDiff between grass and gravel and grass and tarmac surfaces [38]; however, these horses were of a different breed and discipline, and had been ridden in sitting trot.

The relative phase between horse and rider movements captures the overall spatio-temporal organization of the horse-rider system [39] and a smaller time-lag between horse and rider movements may be interpreted as greater synchronisation. The significantly different time offsets around stance on the tarmac and artificial surfaces may reflect differences in surface softness. On the artificial surface, horses' hooves sink more during landing [18] and it may have taken more time for the horses' 'withers and L1' to reach their displacement minima at mid-stance. This delay may have been proportionally larger for the jockey than for the horses because the three 'springs' in this mechanical system must be loaded in series: first the soft surface, followed by the horse limb and lastly the jockey limb. The more unstable soft artificial surface may also initiate reduced coordination within the dyad and lead to a proportionally larger delay for the jockey. In contrast,

reduced time delays for the jockey to follow the horse's movements on tarmac around mid-stance may afford the jockey more time to rise higher during flight. If this is the case, a potential benefit of the jockey rising for longer is that this alleviates pressure on the back of the horse and may allow a greater range of movement in the back [40], without the added load imposed by the jockey. Furthermore, reducing peak pressures around T10-T13 may improve limb kinematics in trot, through increased forelimb and hindlimb protraction and greater carpal and tarsal flexion, and increase thoracolumbar expansion after exercise [41]. However, stability in the seat allows the rider to give effective aids and is crucial for communication between horse and rider [39], and this may be more readily achieved at rising trot-seated (Fig 1). In addition, we observed that the angle of the jockey's back seemed to adopt a more consistent orientation in rising trot compared to two-point seat (S2 Fig in S1 File).

The magnitude of the time offsets around mid-flight were only significantly affected by surface type when the 'withers and L1' landmark was compared to the jockey's pelvis. Here, despite the post-hoc pairwise comparisons not identifying any significantly different surface types, it appeared that the tarmac surface was linked to slightly larger time-offsets when the jockey was doing rising trot; this trend was also apparent at the mid-back and upper-back (Fig 6B). On tarmac, perhaps the horse tended to reach its maximum displacement sooner because it was able to push up more effectively on this level surface offering good grip, while the timing of the jockey's movements was less affected by the hoof-ground interaction. Previous work has identified that when horses better secure their footing, through increased grip at the hoof-surface interface, they experience increased pelvic displacement at push-off [19]. Alternatively, perhaps the jockey rose higher in their stirrups on tarmac, as proposed above, and therefore had a tendency to take longer to reach their maximum displacement point.

### Riding position effects

The differences in time offsets between the mid-stance and mid-flight phases imposed by jockey riding position might reflect a difference in step length in the two halves of the stride cycle. In rising trot, our data indicated that when the jockey was seated in the saddle (seated mid-stance) and then pushed up out of the saddle (post-seated mid-flight), the time taken, as a percentage of the stride, to reach the next stance (standing mid-stance) and flight (post-standing mid-flight) phases was significantly reduced compared to the two-point seat position. This effect was more pronounced between the stance phases, where the offset was 47.8% of the stride cycle for the rising condition, compared to 49.3% in the two-point seat position. Perhaps the additional downwards displacement of the horse's head at 'seated mid-stance', and the reduced withers and sacrum push-off after 'seated mid-stance', limited the 'post-seated mid-flight' duration and caused the horses to place their hooves down sooner for 'standing mid-stance'. However, the biological relevance of this difference requires further investigation.

Previous work evaluating Grand Prix dressage horses ridden in rising trot on a treadmill identified an uneven biphasic load affecting the back, pelvis and limb kinematics, as well as the vertical ground reaction force [42]. That study determined that increased hindlimb loading during sitting stance led to a quicker placement of the hindlimb at sitting stance as there was a shortened protraction: a lower protraction angle (measured in degrees), followed by increased hindlimb retraction (measured in degrees) and increased step length (in m) to accommodate for the increased loading. The authors of that study proposed that horses are urged forward more on the sitting hindlimb through the seat of the rider, thereby increasing the action of the gluteal muscles. For example, if the jockey were riding on the left diagonal, then the horse would be urged forwards after left forelimb and right hindlimb mid-stance; if the jockey were riding on right diagonal, then the horse would be urged forwards after right forelimb and left hindlimb mid-stance. In unridden horses, when there is asymmetric pelvis rise (difference in maximum pelvic position), the side of the horse with reduced vertical displacement has a decreased vertical ground reaction impulse, and is associated with a subsequent increase in horizontal impulse in the second half of the stance [33], which may explain why an increase in step length was previously found after the sitting hindlimb stance [42]. The results reported in our study are consistent with this previous work as the time to the

next stance, or flight, increased after 'standing mid-stance' or 'post-standing mid-flight' (Fig 1), and assuming a constant speed this would imply that there was an increased step length. This observation may be further accentuated by surface conditions. Hoof landing durations are extended on softer surfaces (artificial and grass) relative to firm surfaces (tarmac), which is thought to reflect greater hoof sink on the former [27]. The additional loading imposed by the jockey may serve to cause even greater hoof sink, thereby extending hoof landing durations further and delaying the mid-stance phase and time of upper body vertical displacement minima. Given that our study was predominantly conducted on soft surfaces, there may be an additional effect of increased hoof sink, acting to delay the seated mid-stance and shorten the time to the subsequent standing mid-stance, plus a reduced rate of push-off after the seated mid-stance also shortening the time to the subsequent flight. Indeed, it is worth noting that the offset in our two-point seat data from an exact 50:50 split, could indicate that the horses in our sample had a tendency for motor laterality or sidedness [43] or training-induced asymmetries [44] and this may also have impacted the data.

In our study, we found that the position adopted by the jockey also had a key influence on horse movement asymmetry at both minima (mid-stance) and maxima (mid-flight) phases. During rising trot, jockey riding position induced a significant difference in MinDiff at the poll and withers (although post-hoc pairwise comparisons only found significant differences amongst riding positions at the poll and not at the withers). Specifically, MinDiff values for the left diagonal rising trot and the right diagonal rising trot differed by 14 mm at the poll, and the right diagonal showed 16 mm more negative movement symmetry, relative to the baseline. Movement symmetry is linked to force through Newtonian mechanics [32,33], hence the horses are inferred to have decreased peak vertical force at 'seated mid-stance' through the left forelimb, which had previously been associated with reduced displacement in the 'baseline' condition. However, a more comprehensive assessment of a horse's baseline movement symmetry might be beneficial. The difference between the left and right diagonals may be explained by the position of the jockey in the stride cycle. The downwards action of the jockey's hands during the stance phase may guide the poll in a downwards direction through connection with the reins and bit. Information transfer is a function of the relative motions of the horse and rider [45], and at this point in the stride cycle, the horse and jockey movements are most in sync (Fig 2), likely enhancing haptic information pick-up. In addition, the horse may lower its head more at 'seated mid-stance' because the head is being used to counteract the influence of the jockey; head lowering may aid in flexing the back and thus counteracting the extending moment caused by the rider moving down in the saddle [42]. However, these findings are not in agreement with all previous work, in which poll MinDiff was not found to be significantly affected by rising trot on a straight line [46,47]. Previous work has also identified an increased maximum height reached by the poll after push-off from the forelimb during which stance the rider was sitting, which was attributed to the rider interfering with the normal movement by raising their hands and thus encouraging the horse's head in an upward direction through the bit as the rider rose [46]. In contrast to dressage riders, jockeys appear to adopt a hand position that is more narrow and closer to their horse's neck, which may explain their influence on poll MinDiff rather than MaxDiff in our study. In addition, the shorter stirrup lengths used by jockeys compared to dressage riders may be accountable for some of the differences in findings between the present study and previous work, because changes to stirrup length will alter the position of a rider's centre of mass.

In our study, jockey position was also found to influence push-off asymmetries at the withers and sacrum. Post-hoc comparisons indicated that MaxDiff and UpDiff values were significantly different amongst all three jockey positions ($p < 0.001$, in all cases). For the withers, the right diagonal was associated with positive values, whereas the left diagonal was associated with negative values (Fig 4). Therefore, in rising trot right diagonal, there was reduced push-off after right forelimb (and left hindlimb) stance. Likewise, in rising trot left diagonal, there was reduced push-off after left forelimb (and right hindlimb) stance. This indicated that the jockey may have imposed a withers asymmetry by decreasing the horses' vertical displacement during push-off after 'seated mid-stance'. In addition, it was apparent that, at the withers, there was a significant positive correlation between stride time and MinDiff and between stride time and UpDiff, indicating that the horses became increasingly right-sided asymmetrical with increasing stride time (decreasing speed). Given that all horses

were originally right-sided asymmetrical (or their data flipped such that this was the case), this implies that underlying asymmetries are more apparent at lower speeds and is suggestive of reduced weight-bearing asymmetry with increasing speed; if we assume an increase in speed with decreasing stride time [34]. There was also a significant positive correlation between withers UpDiff and stride time, suggesting there may also have been reduced push-off asymmetry with increasing speed (Fig 5). This is potentially useful for riders/trainers evaluating asymmetries in horses, as they may find it easier to detect movement abnormalities indicative of a lameness when their horses are warming up during ridden exercise at lower speeds. This would then allow them to intervene to remove a horse from competition or training prior to any greater injury being sustained. These observations are consistent with straight-line in-hand trot-up evaluations of mildly lame horses by clinicians, who were found to declare more horses as sound at higher trotting speeds [48].

At the sacrum, negative asymmetry values for the MaxDiff and UpDiff parameters for the 'rising trot right diagonal' indicate there was a left-sided asymmetry in the hind end after left hindlimb (right forelimb) stance (Fig 4). It seems that the addition of the jockey's weight in the preceding seated mid-stance limits the subsequent push-off attainable in both the front and hind end. This seems logical as the horse has to cope with the larger downward impulse from the jockey and then push the jockey up again [42]. Simultaneous force plate and inertial sensor–derived measurements of pelvic displacement and limb loading have indicated that there is an increased fall in the pelvis associated with increased loading at stance [33]. Previous work has also identified decreased pelvic rise when a rider actively rises up in the stirrups, and it was suggested that this rider action creates a downward impulse counteracting the horses push-off [46]. As a result, our data mimic a push-off lameness in the hindlimb that is in stance as the rider sits down in the saddle during the rising trot. The data are also in agreement with the observation of increased downward force through the stirrups when a rider rises from the saddle [40,49]. Considering the poll, withers, and sacrum movements together, it is plausible that the horses employed some compensatory movement patterns. For example, in the left diagonal condition, estimated marginal mean poll and withers MaxDiff values were both negative (left-sided asymmetry), while sacrum values were positive (right-sided asymmetry). This could imply there the horses adapted their hind end movement to compensate for an asymmetry arising in the front end. For the right diagonal condition, estimated marginal mean poll asymmetry values for MaxDiff were negative (left-sided asymmetry), while the withers' MaxDiff were positive (right-sided asymmetry). In the sacrum, the asymmetry values were negative (left-sided asymmetry), which could imply that the horses reacted to the hind end asymmetry by adjusting their front end movement [50,51].

The fact that the movement symmetry data for the two-point seat position do not fall exactly around zero, could reflect an asymmetry in the jockey's movement having an impact on the horses' movement, or that the trotting in-hand on tarmac data did not fully capture the horses' underlying (or 'baseline') asymmetries. It would be interesting to evaluate these relationships in jockeys with varying skill levels, as rider skill has been shown to impact horse motion and phase shifts between rider and horse movements in dressage horse-rider dyads [22,45,52] and it is plausible that horse movement asymmetry could also be influenced. It would also be beneficial to observe the gait kinematics in a larger sample size as the low sample number here is a limitation of the current study. It would also be interesting to evaluate the role of horse age on the data as a yearling or two-year-old might behave differently from an experienced, retired racehorse.

The magnitude of the time offsets between the horse and jockey during the stance phase were found to be affected by jockey riding position. It was apparent that the StanceTDiff–seated was associated with little to no jockey delay compared to the horse (Fig 6A); the StanceTDiff–standing and two-point seat StanceTDiff had a 2.8–4.5% larger time lag, when compared to StanceTDiff–seated, with the exact amount reflecting sensor position on the jockey and surface type (Fig 6A). It is logical for the StanceTDiff–seated parameter to be associated with the smallest time delay between the horse and jockey movements because as the jockey sits they are physically connected to their horse via the saddle. However, as they rise up out of the saddle at StanceTDiff–standing, this position becomes more similar to the two-point seat position, in which horse and jockey movements are decoupled [24] and the delay for the jockey to reach their minimum displacement after the horse increases. The time offsets for StanceTDiff–standing and two-point seat StanceTDiff

appeared most similar for the jockey's pelvis and mid-back (Fig 6A). However, the time offset for the two-point seat position was associated with a slightly greater time offset when the 'withers and L1' and jockey's upper-back were evaluated. This probably reflects the more crouched posture of the jockey in the two-point seat compared to the rising trot position leading to a greater disconnect between the horse at the furthest anatomical position (upper-back) of the jockey.

Jockey position resulted in a significantly different time offset between the jockey and the horse during the flight phases, regardless of which of the three anatomical positions on the jockey were used in the comparison. This was also a much more pronounced effect compared to the influence of surface-type (Fig 6B). It was apparent that both the FlightTDiff–post-seated and two-point seat positions were associated with a negative time offset, indicating that the movement of the jockey lagged behind the horse; this effect was larger for the FlightTDiff–post-seated parameter. In contrast, when the FlightTDiff–post-standing parameter was considered, the movement of the horse lagged behind the jockey (Figs 2 and 6B). There may have been reduced push-off force transferred from the horse to the jockey at StanceTDiff–standing, such that at FlightTDiff–post-standing the jockey movements preceded those of their horse and the jockey reached their displacement maximum sooner but attained a lower maximum when compared to the post-seated flight (Fig 1). When jockey movements precede those of their horse, it may be an effective time point at which a jockey can influence their horse's subsequent movement. However, it is also worth noting that how jockeys perceive different shoe and surface conditions may influence ridden behaviour, such as the time points within a stride cycle during which they communicate with their horse, and resultant movement patterns [53]. In addition, at present it is not possible to evaluate the functional significance of the observed time-offsets between horse and jockey displacement minima and maxima and their bearing on horse-jockey safety.

## Conclusion

This study explored horse kinematics and co-ordination patterns in racehorse-jockey dyads as they trotted over three surface types (tarmac, artificial and grass), with the jockey adopting rising and two-point seat positions. Times between the stance and flight phases of the horse were influenced by jockey riding position, with the standing mid-stance (or post-standing mid-flight) and seated mid-stance (or post-seated mid-flight) rising trot positions becoming closer in time, compared to the two-point seat. In rising trot, the uneven movement of the jockey affected the movement asymmetry of the horse's poll at stance and of the withers and sacrum during push-off. Increased time offsets between horse and jockey movements imply reduced synchronisation, and offsets were increased when the jockey was out of the saddle. Building up an empirical dataset of normative ranges for horse asymmetry and coordination dynamics between jockeys and their horses could enable race training programs to offer feedback to jockeys on how they should adjust their body movements to influence and stabilise their horse's movement. A better understanding of different extrinsic factors influencing the consistency of horse-jockey coordination dynamics at trot is relevant for the ability of horse-jockey dyads to safely commute to gallop tracks and participate in warm-up exercises prior to gallop training and racing over different surfaces.

## Supporting information

**S1 File. Supplementary material.**
(DOCX)

**S2 File. Raw data file.** Raw data also available at: 10.6084/m9.figshare.28890437.
(XLSX)

## Acknowledgments

We would like to thank Lucy Hammond, Lucie Weller, Russell Mackechnie-Guire, Haydn Price and Peter Day for assisting with data collection. The British Racing School are thanked for facilitating access to retired racehorses, a jockey and different racetrack and surface types. We would also like to thank Kieran Kourdache for his comments on the initial study design.

## Author contributions

**Conceptualization:** Kate Horan.

**Data curation:** Kate Horan.

**Formal analysis:** Kate Horan.

**Funding acquisition:** Kate Horan, Thilo Pfau.

**Investigation:** Thilo Pfau.

**Methodology:** Kate Horan, Thilo Pfau.

**Project administration:** Kate Horan, Thilo Pfau.

**Software:** Thilo Pfau.

**Validation:** Kate Horan.

**Visualization:** Kate Horan.

**Writing – original draft:** Kate Horan.

**Writing – review & editing:** Kate Horan, Thilo Pfau.

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
