## [Decision Letter · Decision Letter 0]

9 Jan 2025

PONE-D-24-46468Jockey position and surfaces affect horse movement asymmetry and horse-jockey synchronisation during trotting exercisePLOS ONE

Dear Dr. Horan,

Thank you for submitting your manuscript to PLOS ONE. After careful consideration, we feel that it has merit but does not fully meet PLOS ONE’s publication criteria as it currently stands. Therefore, we invite you to submit a revised version of the manuscript that addresses the points raised during the review process.

We look forward to receiving your revised manuscript.

Kind regards,

Laura Patterson Rosa, M.V., Ph.D.

Academic Editor

PLOS ONE

Journal Requirements:

“This study was funded by a Horserace Betting Levy Board small grant (SPrj049) awarded to KH and TP.”

3. Please ensure that you refer to Figure 3 in your text as, if accepted, production will need this reference to link the reader to the figure.

Reviewers' comments:

Reviewer's Responses to Questions

**Comments to the Author**

1. Is the manuscript technically sound, and do the data support the conclusions?

Reviewer #1: Yes

Reviewer #2: Partly

2. Has the statistical analysis been performed appropriately and rigorously? 

Reviewer #1: Yes

Reviewer #2: Yes

3. Have the authors made all data underlying the findings in their manuscript fully available?

Reviewer #1: Yes

Reviewer #2: No

4. Is the manuscript presented in an intelligible fashion and written in standard English?

Reviewer #1: Yes

Reviewer #2: Yes

5. Review Comments to the Author

Reviewer #1: A really interesting pilot study manuscript that I thought was well written except for a few minor typos throughout. Examples below.

In your abstract, you report "jockeys reduced poll asymmetry", but you only used one jockey.

Last sentence of your introduction: "when the jockey imposes and uneven biphasic load" - did you mean "an"?

Third sentence in "Horse Movement asymmetry": you have an unnecessary comma between "previous study".

Reviewer #2: Dear Authors,

Thank you for submitting your very interesting study. I have taken the liberty of numbering the lines in your manuscript and will refer to them in my comments. I enjoyed reading your paper; however, I found it somewhat lengthy in places, which made it occasionally challenging to follow. I am also somewhat concerned about your sample size, though I acknowledge that including multiple strides may help compensate for this limitation. Please find my comments attached.

6. PLOS authors have the option to publish the peer review history of their article (what does this mean? ). If published, this will include your full peer review and any attached files.

**Do you want your identity to be public for this peer review?** For information about this choice, including consent withdrawal, please see our Privacy Policy .

Reviewer #1: No

Reviewer #2: No

---

## [Author Response · Author response to Decision Letter 0]

1 Mar 2025

Please refer to the attached document for a point-by-point response to all of the reviewer comments.

---

## [Decision Letter · Decision Letter 1]

8 Apr 2025

PONE-D-24-46468R1Effects of jockey position and surfaces on horse movement asymmetry and horse-jockey synchronisation during trotting exercisePLOS ONE

Dear Dr. Horan,

Thank you for submitting your manuscript to PLOS ONE. After careful consideration, we feel that it has merit but does not fully meet PLOS ONE’s publication criteria as it currently stands. Therefore, we invite you to submit a revised version of the manuscript that addresses the points raised during the review process.

We look forward to receiving your revised manuscript.

Kind regards,

Laura Patterson Rosa, M.V., Ph.D.

Academic Editor

PLOS ONE

Journal Requirements:

Reviewers' comments:

Reviewer's Responses to Questions

**Comments to the Author**

1. If the authors have adequately addressed your comments raised in a previous round of review and you feel that this manuscript is now acceptable for publication, you may indicate that here to bypass the “Comments to the Author” section, enter your conflict of interest statement in the “Confidential to Editor” section, and submit your "Accept" recommendation.

Reviewer #1: (No Response)

Reviewer #2: All comments have been addressed

2. Is the manuscript technically sound, and do the data support the conclusions?

Reviewer #1: Yes

Reviewer #2: Yes

3. Has the statistical analysis been performed appropriately and rigorously? 

Reviewer #1: Yes

Reviewer #2: Yes

4. Have the authors made all data underlying the findings in their manuscript fully available?

Reviewer #1: No

Reviewer #2: Yes

5. Is the manuscript presented in an intelligible fashion and written in standard English?

Reviewer #1: Yes

Reviewer #2: Yes

6. Review Comments to the Author

Reviewer #1: The authors have addressed my previous comments, and the manuscript is much easier to read and follow. However, the location for accessing the raw data is not provided. Only summary statistics are provided in the supplemental tables.

In Line 111, the authors mention the operating system for the laptop, but don't mention the computer specifications (memory, storage, processor, etc.).

Reviewer #2: Thank you for providing detailed responses to my comments. Your manuscript reads really well and is now fully suitable for publication.

7. PLOS authors have the option to publish the peer review history of their article (what does this mean? ). If published, this will include your full peer review and any attached files.

**Do you want your identity to be public for this peer review?** For information about this choice, including consent withdrawal, please see our Privacy Policy .

Reviewer #1: No

Reviewer #2: No

---

## [Author Response · Author response to Decision Letter 1]

29 Apr 2025

We have updated details about the laptop as requested by reviewer 2 and created a data file that is available on FigShare. We provide a link to the data in the revised manuscript.

All other changes marked are just corrections for typos.

---

## [Editor Report · Decision Letter 2]

2 May 2025

Effects of jockey position and surfaces on horse movement asymmetry and horse-jockey synchronisation during trotting exercise

PONE-D-24-46468R2

Dear Dr. Horan,

We’re pleased to inform you that your manuscript has been judged scientifically suitable for publication and will be formally accepted for publication once it meets all outstanding technical requirements.

Kind regards,

Laura Patterson Rosa, M.V., Ph.D.

Academic Editor

PLOS ONE
---

## [Editor Report · Acceptance letter]

PONE-D-24-46468R2

PLOS ONE

Dear Dr. Horan,

I'm pleased to inform you that your manuscript has been deemed suitable for publication in PLOS ONE. Congratulations! Your manuscript is now being handed over to our production team.

Kind regards,

on behalf of

Dr. Laura Patterson Rosa

Academic Editor

PLOS ONE